

# Front dynamics in the XY chain after local excitations

**Viktor Eisler and Florian Maislinger**

Institut für Theoretische Physik, Technische Universität Graz,
Petersgasse 16, A-8010 Graz, Austria

⋆ viktor.eisler@tugraz.at

## Abstract

We study the time evolution of magnetization and entanglement for initial states with local excitations, created upon the ferromagnetic ground state of the XY chain. For excitations corresponding to a single or two well separated domain walls, the magnetization profile has a simple hydrodynamic limit, which has a standard interpretation in terms of quasiparticles. In contrast, for a spin-flip we obtain an interference term, which has to do with the nonlocality of the excitation in the fermionic basis. Surprisingly, for the single domain wall the hydrodynamic limit of the entropy and magnetization profiles are found to be directly related. Furthermore, the entropy profile is additive for the double domain wall, whereas in case of the spin-flip excitation one has a nontrivial behaviour.


# 1  Introduction

The nonequilibrium dynamics of integrable quantum many-body systems has been the focus of intensive research [1]. The interest in these peculiar models, characterized by the existence of a large set of conservation laws, comes from two main perspectives. On one hand, they show relaxation towards generalized stationary ensembles that are not described by conventional statistical mechanics [2]. On the other hand, owing to the presence of stable quasiparticle excitations, integrable models have anomalous transport properties [3]. A recent milestone in understanding the transport driven by an initial inhomogeneity has been the formulation of generalized hydrodynamics (GHD) [4,5], which gives accurate predictions for the profiles of conserved densities in an appropriate spacetime scaling limit.

   The simplest paradigm of an inhomogeneous initial state is a domain wall, separating domains of spins with different magnetizations. Letting the system evolve, the domain wall starts to melt, giving rise to an expanding front region characterized by a nonzero spin current. The resulting magnetization profiles were studied in various integrable spin models such as the XX chain [6–8], the transverse Ising (TI) [9–11], the XY [12] as well as the XXZ chains [4,13–17]. Rather generically one finds ballistic transport, with the exception of the isotropic Heisenberg chain where a diffusive behaviour is observed instead [18–23]. The common feature in all of the examples above is that the domain wall is oriented along the $z$-axis, and thus the magnetization is a local operator in the fermionic representation of the corresponding spin chain. In particular, for models with fermion-number conservation, the transverse magnetization itself corresponds to a locally conserved density, which makes the problem directly amenable to GHD techniques.

   Recently, however, domain walls created upon the symmetry-broken ferromagnetic ground states of TI or XY chains have been considered [24–26]. The ordering in these chains occurs in the longitudinal component of the magnetization, which is a highly nonlocal string operator in the fermionic picture, being nontrivially related to the local conserved densities. Hence, even though one has a free-fermion model at hand, it is a priori unclear whether a hydrodynamic description still holds for this observable. Nevertheless, in [25,26] it has been shown that, for domain walls excited by a single local fermion operator, the longitudinal magnetization profile has the usual hydrodynamic scaling limit one would naively expect. Namely, the profile is determined by noninteracting quasiparticles carrying the fraction of a spin-flip and traveling at the corresponding group velocity.

   In the present work we extend these studies to excitations that can be written as the product of two local fermion operators. In the spin language they describe a double domain wall, and if the distance between them is sufficiently large, we find that the magnetization profile factorizes in the hydrodynamic scaling limit. In other words, the quasiparticle excitations created at the two domain walls are completely independent. In contrast, the situation becomes nontrivial if the fermionic excitations act on neighbouring sites, even though the product of two adjacent domain walls is just a spin-flip and thus perfectly local in the spin-representation. Indeed, it turns out that this composite fermionic excitation leads to interference effects between the

quasiparticle modes, encoded in the form factors of the spin operator. This interference term yields a significant contribution to the hydrodynamic profile, which can be found analytically via stationary phase analysis.

We also study in detail the correlation functions and the entanglement entropy for the single domain wall excitation. Interestingly, both of them can be directly related to the magnetization. For the correlations we derive a relation which holds also for finite times if the separation of the spins is much larger than the correlation length. On the other hand, for the entropy we propose an ansatz that is motivated by recent results for single-mode quasiparticle excitations in a free massive quantum field theory (QFT) [27,28]. Our ansatz works perfectly in the hydrodynamic regime, thereby creating an exact relation between the magnetization and entanglement profiles. Furthermore, we observe that the entropy becomes additive for the double domain wall excitation, whereas for the spin-flip one has again a nontrivial behaviour due to the above mentioned interference terms.

The paper is structured as follows. We start by introducing the model in Sec. 2. The magnetization dynamics is studied in Sec. 3 for three different local excitations as well as for a local quench. The correlation functions are investigated in Sec. 4, followed by the study of the entropy profiles in Sec. 5. We discuss our findings in Sec. 6, and the technical details of the calculations are reported in three Appendices.

## 2 Model

We consider an XY spin chain of length $N$ described by the Hamiltonian

$$H = -\sum_{n=1}^{N-1}\left(\frac{1+\gamma}{4}\sigma_n^x\sigma_{n+1}^x + \frac{1-\gamma}{4}\sigma_n^y\sigma_{n+1}^y\right) - \frac{h}{2}\sum_{n=1}^{N}\sigma_n^z, \tag{1}$$

where $\sigma_n^\alpha$ are Pauli matrices located at site $n$, $h$ and $\gamma$ denote the transverse magnetic field and the XY anisotropy, respectively. We restrict ourselves to the parameter regime $0 < h < 1$ and $0 < \gamma \le 1$ where the chain is in a gapped ferromagnetic phase, with $\gamma = 1$ corresponding to the TI chain.

The Hamiltonian (1) is diagonalized through a standard procedure [29], by first introducing Majorana fermions via a Jordan-Wigner transformation

$$a_{2j-1} = \prod_{k=1}^{j-1}\sigma_k^z\sigma_j^x, \qquad a_{2j} = \prod_{k=1}^{j-1}\sigma_k^z\sigma_j^y, \tag{2}$$

satisfying anticommutation relations $\{a_k, a_l\} = 2\delta_{k,l}$. While (1) describes an open chain which is most suitable for our numerical calculations, the analytical treatment of the problem requires to consider either periodic ($s = +$) or antiperiodic ($s = -$) boundary conditions, $\sigma_{N+1}^x = s\sigma_1^x$ and $\sigma_{N+1}^y = s\sigma_1^y$. Due to the global spin-flip symmetry of the model, the corresponding Hamiltonians can then be split into two parts

$$H_s = \frac{1-s\mathcal{P}}{2}H_R + \frac{1+s\mathcal{P}}{2}H_{NS}, \qquad \mathcal{P} = \prod_{n=1}^{N}\sigma_n^z. \tag{3}$$

In terms of the Majorana fermions, the corresponding symmetry sectors are described by the Hamiltonians

$$H_{R/NS} = \frac{i}{2}\sum_{j=1}^{N}\left(\frac{1+\gamma}{2}a_{2j}a_{2j+1} - \frac{1-\gamma}{2}a_{2j-1}a_{2j+2} + ha_{2j-1}a_{2j}\right), \tag{4}$$

which differ in the boundary conditions $a_{2N+1} = \pm a_1$ and $a_{2N+2} = \pm a_2$ being periodic for the Ramond (R) and antiperiodic for the Neveu-Schwarz (NS) sectors.

In order to diagonalize (4), one performs a Fourier transformation followed by a Bogoliubov rotation

$$
\begin{aligned}
a_{2j-1} &= \frac{1}{\sqrt{N}} \sum_{q \in \text{R/NS}} e^{-iqj} e^{i(\theta_q + q)/2} (b_q^\dagger + b_{-q}), \\
a_{2j} &= \frac{-i}{\sqrt{N}} \sum_{q \in \text{R/NS}} e^{-iqj} e^{-i(\theta_q + q)/2} (b_q^\dagger - b_{-q}),
\end{aligned}
\tag{5}
$$

where the Bogoliubov angle and the dispersion are given by

$$
e^{i(\theta_q + q)} = \frac{\cos q - h + i\gamma \sin q}{\epsilon_q}, \qquad \epsilon_q = \sqrt{(\cos q - h)^2 + \gamma^2 \sin^2 q}. \tag{6}
$$

Note that the above definition ensures that the function $\theta_q$ is continuous within the Brillouin zone $q \in [-\pi, \pi]$. To satisfy the proper boundary conditions, the allowed values of the momenta are $q_k = \frac{2\pi}{N}k$ for R and $q_k = \frac{2\pi}{N}(k+1/2)$ for NS, respectively, with $k = -N/2, \dots, N/2-1$ and $N$ even. The diagonalized Hamiltonian and its $K$-particle eigenstates are then given by

$$
H_{\text{R/NS}} = \sum_{q \in \text{R/NS}} \epsilon_q b_q^\dagger b_q + \text{const}, \qquad |q_1, q_2, \dots, q_K\rangle_{\text{R/NS}} = \prod_{i=1}^{K} b_{q_i}^\dagger |0\rangle_{\text{R/NS}}. \tag{7}
$$

It should be stressed that the eigenstates with $K$ even belong to the spin-periodic Hamiltonian $H_+$, whereas the eigenstates of the spin-antiperiodic $H_-$ have odd $K$.

In the thermodynamic limit $N \to \infty$, the periodic chain $H_+$ has a doubly degenerate ground state with ferromagnetic ordering along the $x$-axis, denoted by $|\Uparrow\rangle$ and $|\Downarrow\rangle$, respectively. Note however, that for finite $N$ the actual ground states in both symmetry sectors are given by

$$
|0\rangle_{\text{NS}} = \frac{1}{\sqrt{2}}(|\Uparrow\rangle + |\Downarrow\rangle), \qquad |0\rangle_{\text{R}} = \frac{1}{\sqrt{2}}(|\Uparrow\rangle - |\Downarrow\rangle), \tag{8}
$$

which are separated by an exponentially small gap and both have vanishing magnetizations.

## 3 Magnetization dynamics

We are interested in the dynamics of the magnetization of various initial states, excited locally from the ferromagnetic ground state $|\Uparrow\rangle$ and time-evolved under the Hamiltonian $H$ in (1). The locality of the excitation is understood in terms of the Majorana basis, which implies that these excitations may become highly non-local in the spin-basis representation. In fact, the latter will correspond to domain-wall excitations and one is interested in how the inhomogeneity spreads out under unitary time evolution. On the other hand, since the order-parameter magnetization is not conserved, even a single spin-flip excitation (which is local in terms of the spins) will lead to nontrivial dynamics. For the study of domain-wall melting, we will also consider for comparison a local quench setup where two separate chains are initially prepared in oppositely magnetized ground states, and subsequently joined together.

The time-evolved magnetization can be extracted in a number of different ways. On the numerical side, we apply matrix product state (MPS) calculations[1] [30] in an open-chain geometry. To ensure that we obtain the proper ferromagnetic (symmetry-broken) ground state $|\Uparrow\rangle$, we introduced a small longitudinal field $h_x > 0$ in the Hamiltonian $H - h_x \sum_i \sigma_i^x$ for the

---

[1]Our MPS code is implemented using the ITENSOR library, http://itensor.org/.

first few sweeps and set $h_x = 0$ afterwards, until convergence is reached. The excitations are then created by acting with the matrix product operator representation of the corresponding spin-excitation. Finally, the time evolution was implemented with the finite two-site time-dependent variational principle (TDVP) algorithm [31].

On the other hand, we also employed Pfaffian techniques for the numerical evaluation of the magnetization. For the simple domain-wall excitation these were described in Ref. [25], but the calculations can easily be generalized for the other local excitations we deal with. In all of the examples we observed a perfect agreement with the results of MPS calculations.

Finally, we also present analytical results based on form-factor calculations. To this end, one has to first express the excited initial state $|\psi_0\rangle = (|\psi_0\rangle_{\mathrm{R}} + |\psi_0\rangle_{\mathrm{NS}})/\sqrt{2}$ in the fermion basis, which is then time-evolved with the corresponding Hamiltonian in both symmetry sectors as

$$|\psi_t\rangle_{\mathrm{R/NS}} = \mathrm{e}^{-itH_{\mathrm{R/NS}}}|\psi_0\rangle_{\mathrm{R/NS}}. \tag{9}$$

Once $|\psi_0\rangle_{\mathrm{R/NS}}$ is written as a linear combination of the $K$-particle eigenstates (7), the time evolution is trivial

$$\mathrm{e}^{-itH_{\mathrm{R/NS}}}|q_1, q_2, \ldots, q_K\rangle_{\mathrm{R/NS}} = \mathrm{e}^{-it\sum_{k=1}^K \epsilon_{q_k}}|q_1, q_2, \ldots, q_K\rangle_{\mathrm{R/NS}}, \tag{10}$$

since the Hamiltonian $H_{\mathrm{R/NS}}$ is diagonal in this basis. It is useful to introduce the normalized magnetization which can be evaluated as

$$\mathcal{M}_n(t) = \frac{{}_{\mathrm{R}}\langle\psi_t|\sigma_n^x|\psi_t\rangle_{\mathrm{NS}}}{{}_{\mathrm{R}}\langle0|\sigma_n^x|0\rangle_{\mathrm{NS}}}. \tag{11}$$

Note that, since the operator $\sigma_n^x$ changes the parity of the state, the only non-vanishing contribution to the expectation value is between different parity sectors. In turn, the calculation of $\mathcal{M}_n(t)$ boils down to evaluating multiple sums over the momenta with the form factors ${}_{\mathrm{R}}\langle p_1, \ldots, p_L|\sigma_n^x|q_1, \ldots, q_K\rangle_{\mathrm{NS}}$, which are known explicitly from previous studies [32–34]. In the following we always consider the thermodynamic limit $N \to \infty$, where the sums over momenta can be turned into integrals and the expressions for the form factors are summarized in Appendix A.

### 3.1 Single domain wall

Our first example is a single domain wall, which has already been considered for the TI [25] as well as for the XY chains [26]. For completeness, we revisit here the results obtained previously for the normalized magnetization. The single domain wall is an excitation $|\psi_0\rangle = D_{n_1}|\Uparrow\rangle$ created by the operator

$$D_{n_1} = \prod_{j=1}^{n_1-1} \sigma_j^z \sigma_{n_1}^x = a_{2n_1-1}. \tag{12}$$

As remarked before, $D_{n_1}$ is strictly local in terms of the fermions, whereas in the spin representation it creates spin-flips all over the sites $j < n_1$. In the eigenbasis of the Hamiltonian it corresponds to a linear combination of one-particle states

$$|\psi_0\rangle = \frac{1}{\sqrt{N}}\sum_q \mathrm{e}^{-iq(n_1-1/2)}\mathrm{e}^{i\theta_q/2}|q\rangle, \tag{13}$$

where we have suppressed the subscripts R/NS of the symmetry sector for notational simplicity. One thus only needs the form factors between one-particle states, which has a relatively simple

form (51) given in Appendix A. Performing the time evolution (9) via (10) and inserting the result into (11), one arrives at

$$\mathcal{M}_n(t) = \int_{-\pi}^{\pi} \frac{dp}{2\pi} \int_{-\pi}^{\pi} \frac{dq}{2\pi} \frac{\epsilon_p + \epsilon_q}{2\sqrt{\epsilon_p \epsilon_q}} \frac{e^{i(n-n_1+1/2)(q-p)}}{i \sin\left(\frac{q-p}{2}\right)} e^{i(\theta_q - \theta_p)/2} e^{-i(\epsilon_q - \epsilon_p)t} . \tag{14}$$

The above expression simplifies considerably in appropriate scaling limits. Indeed, noting that the integral receives the dominant contribution due to a pole at $q = p$ in the integrand of (14), one can change variables as $Q = q - p$ and $P = (q + p)/2$, and perform a stationary phase analysis as described in Appendix B. In turn, one obtains

$$\mathcal{M}_n(t) = 1 - 2 \int_{-\pi}^{\pi} \frac{dP}{2\pi} \Theta(v_P - v), \qquad v = \frac{n - n_1 + 1/2}{t}, \tag{15}$$

which is the so-called hydrodynamic scaling limit. Here $\Theta(x)$ is the Heaviside step function, $v_P = \frac{d\epsilon_P}{dP}$ is the group velocity of the single-particle excitations and $v$ is the ray variable, with the distance measured from the initial location $n_1 - 1/2$ of the domain wall. The result (15) has a simple semiclassical interpretation, which has been applied many times to understand front dynamics in quantum chains [35–38]. Namely, the magnetization is transported by single-particle excitations, each carrying an elementary spin-flip, which contribute to the hydrodynamic profile at a given ray only if their velocity $v_P > v$.

Another interesting scaling regime emerges around the edge of the front $v \approx v_{max}$, given by the maximum speed of excitations. In order to understand the fine structure of the edge, a higher order stationary phase analysis has to be performed around the momentum $q_*$ which yields the maximum velocity $v_{q_*} = v_{max}$. As shown in Appendix B, this leads to the following result

$$\mathcal{M}_n(t) \approx 1 - 2 \left(\frac{2}{|v''_{q_*}|t}\right)^{1/3} \rho(X), \qquad X = (n - n_1 + 1/2 + \theta'_{q_*}/2 - v_{q_*}t) \left(\frac{2}{|v''_{q_*}|t}\right)^{1/3}. \tag{16}$$

In other words, with the proper choice of the scaling variable $X$ measuring the distance from the edge, and after appropriate rescaling, the fine structure of the magnetization front is given via the function

$$\rho(X) = \mathcal{K}_{Ai}(X, X) = \left[\text{Ai}'(X)\right]^2 - X\,\text{Ai}^2(X). \tag{17}$$

Note that $\rho(X)$ is nothing else but the diagonal part of the Airy-kernel $\mathcal{K}_{Ai}(X, Y)$ [39], which appears in a number of front evolution problems related to free-fermion edge universality [8, 11, 40–45].

The results (15) and (16) have already been tested against numerical calculations for various parameters of the XY chain, where the notable feature of a hydrodynamic phase transition at $h_c = 1 - \gamma^2$ was observed [26]. Indeed, this phase transition can be understood by the appearance of a second local maximum in the group velocities $v_q$ for $h < h_c$, which in turn leads to kinks in the bulk of the hydrodynamic magnetization profile [26].

Finally, it should be noted that the analytical result was obtained by following the time evolution of one-particle states building up the domain wall. Strictly speaking, these states are eigenstates of $H_-$ only, i.e. the time evolution has to be performed with antiperiodic boundary conditions on the spin chain. However, since the form factor calculations are carried out directly in the thermodynamic limit, the boundaries actually do not play any role.

## 3.2 Double domain wall

We now move on to consider more complicated excitations, that are created by acting with the operator

$$D_{n_1,n_2} = \sigma^x_{n_1-1} \prod_{j=n_1}^{n_2-1} \sigma^z_j \, \sigma^x_{n_2} = -i \, a_{2n_1-2} \, a_{2n_2-1} \,, \tag{18}$$

where $n_2 > n_1$ is assumed. In terms of fermions this is a two-local operator, i.e. supported on two sites only. In contrast, $D_{n_1,n_2}$ is again nonlocal in the spin representation, and it is easy to see that it describes a double domain wall, located at sites $n_1$ and $n_2$, respectively. Using (5), the excited initial state can be written as

$$|\psi_0\rangle = \frac{1}{N}\sum_q e^{iq(n_2-n_1)}e^{-i\theta_q}|0\rangle - \frac{1}{N}\sum_{q_1,q_2} e^{-iq_1(n_1-1/2)}e^{-iq_2(n_2-1/2)}e^{-i(\theta_{q_1}-\theta_{q_2})/2}|q_1,q_2\rangle \,. \tag{19}$$

We shall restrict ourselves to the case $n_2-n_1 \gg 1$, i.e. when the two domain walls are spatially well separated, such that the sum in the first term of (19) becomes highly oscillatory and can be neglected. The initial state then involves only two-particle excitations and the time evolved state can be written as

$$|\psi_t\rangle = -\frac{1}{N}\sum_{q_1,q_2} e^{-iq_1(n_1-1/2)}e^{-iq_2(n_2-1/2)}e^{-i(\theta_{q_1}-\theta_{q_2})/2}e^{-i(\epsilon_{q_1}+\epsilon_{q_2})t}|q_1,q_2\rangle \,. \tag{20}$$

The magnetization $\mathcal{M}_n(t)$ can thus be expressed as a quadruple integral via two-particle form factors ${}_R\langle p_1,p_2|\sigma^x_n|q_1,q_2\rangle_{NS}$, that are reported in (53) in Appendix A. The result can be simplified, similarly to the single domain wall case, by analyzing the pole-structure of the form factors combined with a stationary phase approximation. The poles appear for momenta satisfying $q_1 = p_1$ and $q_2 = p_2$ or $q_1 = p_2$ and $q_2 = p_1$. For the first pole one obtains two independent stationary phase conditions

$$v_{P_i} t - (-1)^i \theta'_{P_i} - (n - n_i + 1/2) = 0, \tag{21}$$

where $P_i = (q_i + p_i)/2$ for $i = 1,2$. Note that this pole corresponds to a process where the incoming momenta are matched with the outgoing ones at each domain wall separately. In contrast, at the second pole an incoming momentum of the first domain wall must match with an outgoing momentum of the second domain wall. However, as shown in Appendix B, after the exchange of the outgoing momenta and under the assumption $n_2-n_1 \gg 1$, the stationary phase condition cannot be satisfied. Thus only the first pole gives a contribution to the integral and leads to the result

$$\mathcal{M}_n(t) = \prod_i \int \frac{dP_i}{2\pi}\big[1 - 2\Theta\big(v_{P_i} - v_i\big)\big], \qquad v_i = \frac{n - n_i + 1/2}{t}\,. \tag{22}$$

The hydrodynamic scaling limit of the profile in (22) has thus a factorized form with again a very simple physical interpretation. The ray variables $v_i$ now measure the distances from the corresponding initial domain wall locations $n_i - 1/2$, where quasiparticles with velocity $v_{P_i}$ are emitted, each carrying a spin-flip. If, for a given pair of particles, one has $v_{P_1} > v_1$ and $v_{P_2} > v_2$ then both of the particles have reached site $n$ at time $t$, hence the spin is flipped twice and one has a positive contribution. If, on the other hand, $v_{P_1} < v_1$ and $v_{P_2} > v_2$, then only one particle has arrived and the contribution is negative. The profile is then obtained by summing the contributions over all pairs.

In Fig. 1 we show the results of our MPS simulations together with the result (22). One can see a perfect agreement, even after the two fronts propagating from different locations overlap

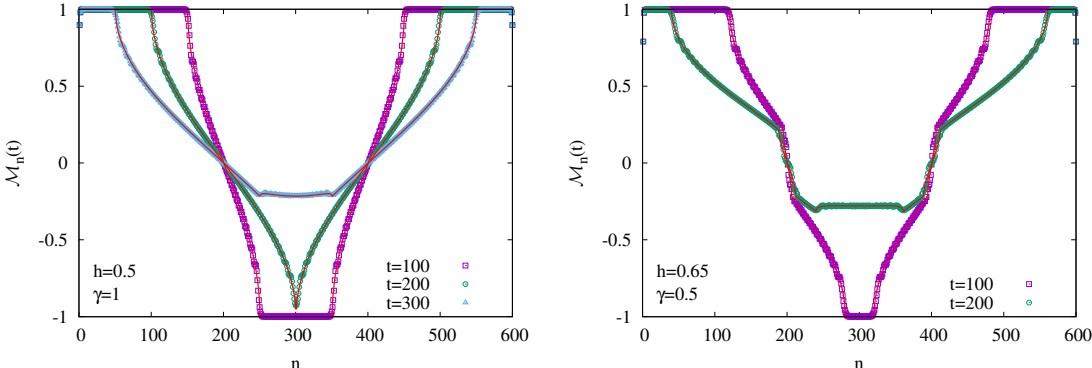

Figure 1: Magnetization profiles after a double domain wall excitation for different times and various $h$ and $\gamma$. The solid red lines show the approximation (22). The parameters are $N = 600$, $n_1 = 201$ and $n_2 = 401$.

in the middle. In particular, one observes the emergence of two cusps at the ends of the overlap region, which follows from the factorized form of (22), i.e. one multiplies two single domain wall front profiles, each having square-root singularities at their edges. Moreover, this also implies that the outer edge of the front is still described by the same scaling (16) as for the single domain wall. On the right of Fig. 1 there are extra kinks to be seen, which is due to the fact that one has $h < h_c$ there, i.e. one is beyond the hydrodynamical phase transition point.

## 3.3 Single spin-flip

After having discussed the evolution of domain walls, we now study a very simple excitation, in the form of a single flipped spin. Naively, one would think that this excitation has a trivial hydrodynamic limit, and the flipped spin just disperses. However, since the magnetization is not conserved under the XY dynamics, it turns out that the profile is far from being trivial. In fact, the operator that creates a spin-flip at site $n_1$ is just $\sigma_{n_1}^z = -i a_{2n_1-1} a_{2n_1}$, which is strictly local in the spin representation, but is again two-local, i.e. a product of two adjacent Majoranas in the fermionic picture. Hence, this form is more reminiscent of a double domain wall excitation, with the exception that they are now created at neighbouring sites. Rewriting the excitation in the fermionic basis one has

$$|\psi_0\rangle = m^z |0\rangle - \frac{1}{N} \sum_{q_1,q_2} e^{-iq_1(n_1-1/2)} e^{-iq_2(n_1+1/2)} e^{i(\theta_{q_1}-\theta_{q_2})/2} |q_1,q_2\rangle , \qquad (23)$$

where the ground-state contribution is now proportional to the transverse magnetization

$$m^z = \langle 0|\sigma_n^z|0\rangle = -\int_{-\pi}^{\pi} \frac{dq}{2\pi} e^{i(\theta_q+q)} , \qquad (24)$$

and thus cannot be neglected.

The calculation of $\mathcal{M}_n(t)$ follows the same steps as in the previous cases. Note, in particular, that the two-particle contribution in (23) has almost the same form as (19) for the double domain wall with $n_2 = n_1 + 1$, except for the sign of the Bogoliubov phases. After time evolving and taking the expectation value with $|\psi_t\rangle$, one has now cross terms where the form factors $_R\langle 0|\sigma_n^x|q_1,q_2\rangle_{NS}$ appear, see (52). However, since they have no poles, it is easy to see that their contribution is negligible in the scaling limit we are interested in. On the other hand, the two-particle form factors now yield a contribution from both of the poles. Indeed, the stationarity condition is, up to the sign of the $\theta'_{P_i}$ term, is the same as (21) for the double

domain wall with $n_2 = n_1 + 1$. However, in the limit of $t \gg 1$ and $|n - n_1| \gg 1$, the two equations are essentially the same. Hence, the process in which an incoming momentum of the first domain wall scatters into an outgoing momentum of the neighbouring one is equally well permitted and yields a sizable contribution.

Carrying out the stationary phase analysis in detail (see Appendix B), one arrives at the following result in the hydrodynamic limit

$$\mathcal{M}_n(t) = (m^z)^2 + \left[1 - 2\int_{-\pi}^{\pi} \frac{dP}{2\pi} \Theta(v_P - \tilde{v})\right]^2 - \left|m^z + 2\int_{-\pi}^{\pi} \frac{dP}{2\pi} e^{iP} e^{i\theta_P} \Theta(v_P - \tilde{v})\right|^2, \quad (25)$$

where the ray variable $\tilde{v} = \frac{n - n_1}{t}$ is slightly changed compared to (15), since the distance is now measured from the location $n_1$ of the spin-flip. The profile can be written as the sum of three terms, where the first one is simply the ground-state contribution. The second one corresponds to the factorized result for the double domain wall and the third one describes a kind of interference term, where the momenta of the excitations building up the two domain walls are exchanged. There is no simple semiclassical interpretation of this interference term, since the quasiparticles contribute with a phase factor. The result (25) is compared against our numerical calculations in Fig. 2 with an excellent agreement.

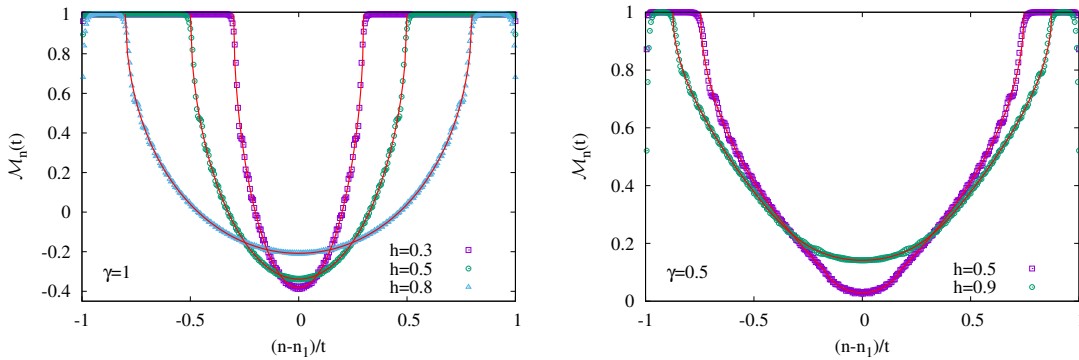

Figure 2: Magnetization profiles after a spin-flip excitation for various $h$ and $\gamma$. The red solid lines show the approximation (25). The parameters are $N = 400$, $n_1 = 200$ and $t = 200$.

It is also interesting to have a look at the edge behaviour of the profile. Performing the higher order stationary phase analysis (see Appendix B), one is led to the following result

$$\mathcal{M}_n(t) \approx 1 - 2\left(\frac{2}{|v_{q_*}''|t}\right)^{1/3} \tilde{\rho}(\tilde{X}), \qquad \tilde{X} = (n - n_1 - v_{q_*}t)\left(\frac{2}{|v_{q_*}''|t}\right)^{1/3}, \quad (26)$$

where the scaling function is given by

$$\tilde{\rho}(\tilde{X}) = \left[2 + 2m^z \cos(\theta_{q_*} + q_*)\right] \mathcal{K}_{Ai}(\tilde{X}, \tilde{X}). \quad (27)$$

The result is thus very similar to the one for the domain wall in (16), however the scaling function $\tilde{\rho}(\tilde{X})$ acquires a nontrivial prefactor, which depends explicitly on the transverse magnetization $m^z$, and even on the Bogoliubov phase evaluated at $q_*$ where the quasiparticle velocity has its maximum. In particular, this phase factor vanishes for the TI chain and one has a factor of 2 difference with respect to $\rho(X)$. This explains the numerical findings of Ref. [24] where the very same setup was studied. We checked the validity of the edge scaling (26) in Fig. 3 for various parameter values and found a very good agreement, there are however some differences in the convergence towards the scaling function $\tilde{\rho}(\tilde{X})$.

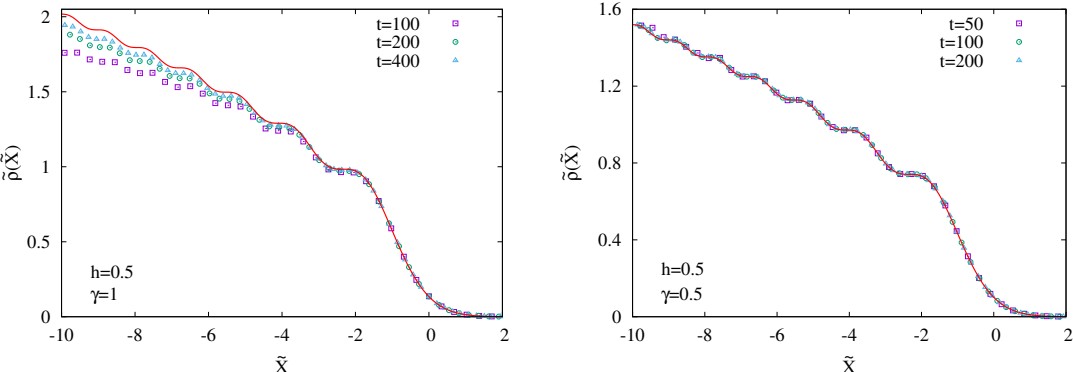

Figure 3: Edge scaling (26) of the magnetization profiles after a spin-flip excitation for various $h$ and $\gamma$. The red solid lines show the scaling function in (27).

## 3.4 Local quench

As a final example, we show here the results for the magnetization profile resulting from a local quench. That is, instead of applying a local excitation to the symmetry-broken ferromagnetic state, we rather prepare the two halves of our chain in oppositely magnetized ground states and join them together. Our goal is to check whether this protocol yields a similar result for the hydrodynamic profile as the one found for the single domain wall excitation.

The initial and time-evolved states are now given by

$$|\psi_0\rangle = |\Downarrow\rangle \otimes |\Uparrow\rangle, \qquad |\psi_t\rangle = \mathrm{e}^{-iHt}|\psi_0\rangle. \tag{28}$$

Since our initial state is not prepared as an excitation upon the bulk vacuum state, it is a nontrivial question how $|\psi_0\rangle$ can be written in the basis of the full Hamiltonian $H$. Thus we shall only perform numerical (MPS and Pfaffian based) calculations for the quench. The results, shown in Fig. 4, turn out to be rather surprising. Namely, we find that in the TI limit ($\gamma = 1$) the profiles after the local quench (full symbols) almost exactly coincide with the ones for the domain wall excitation (empty symbols). The only deviations visible at the scale of the figure are around the front edges. In sharp contrast, for $\gamma = 0.5$ one has a huge deviation between the profiles for all the values of $h$ we considered. This signals that in the latter case the factorized initial state is not well approximated by a single-particle excitation in the fermionic basis. We observe that the mismatch between the profiles gradually increases as one moves away from the TI limit. However, we have no clear explanation of this phenomenon which needs further studies.

# 4 Correlation functions

The form-factor approach is not restricted to the study of the magnetization profile. The next simplest physically interesting observable is the correlation function between the spins. Here we shall restrict ourselves to equal-time correlations between the $x$-components of the spin, which have already been addressed briefly in [26]. It is useful to work with the normalized correlation functions

$$\mathcal{C}_{m,n}(t) = {}_{\mathrm{NS}}\langle\psi_t|\hat{\mathcal{M}}_m\hat{\mathcal{M}}_n|\psi_t\rangle_{\mathrm{NS}}, \tag{29}$$

where the expectation value is now taken between the NS components only, since the operator $\sigma_m^x\sigma_n^x$ does not change the parity. Note that we use here that the corresponding expectation value between the R components is equal to (29) in the thermodynamic limit.

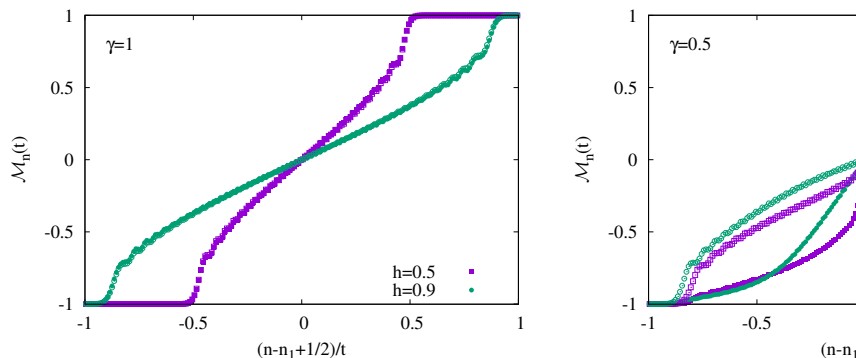

Figure 4: Magnetization profiles after the local quench (full symbols) vs. single domain wall excitation (empty symbols), for various $h$ and $\gamma$. The parameters are $N = 400$, $n_1 = 201$ and $t = 100$.

In order to get a form-factor expansion of (29), we shall insert the resolution of the identity

$$\mathbb{1} = |0\rangle \langle 0| + \sum_p |p\rangle \langle p| + \sum_{p_1,p_2} |p_1, p_2\rangle \langle p_1, p_2| + \sum_{p_1,p_2,p_3} |p_1, p_2, p_3\rangle \langle p_1, p_2, p_3| + \dots \qquad (30)$$

Note that the resolution must be taken within the R sector, but we omit here the subscripts for notational simplicity. The form-factor expansion can be obtained by inserting the expression of $|\psi_t\rangle_{\text{NS}}$ in terms of the fermionic basis. We focus here on the case of a single domain wall, since the calculations become rather cumbersome for more complicated excitations. In this case $|\psi_t\rangle_{\text{NS}}$ is a superposition of single-particle states only and it is reasonable to assume that, for distances much larger than the correlation length $|n - m| \gg \xi$, the dominant contribution to the correlations comes from the single-particle terms in (30) as well. To lowest order in the form-factor expansion we thus arrive at the result

$$\mathcal{C}_{m,n}(t) \simeq \int \frac{dq_1}{2\pi} \int \frac{dq_2}{2\pi} e^{-i(\theta_{q_1} - \theta_{q_2})/2} e^{i(\epsilon_{q_1} - \epsilon_{q_2})t}$$
$$\times \int \frac{dp}{2\pi} \frac{\epsilon_p + \epsilon_{q_1}}{2\sqrt{\epsilon_p \epsilon_{q_1}}} \frac{\epsilon_p + \epsilon_{q_2}}{2\sqrt{\epsilon_p \epsilon_{q_2}}} \frac{e^{-i(m-n_1+1/2)(q_1-p)}}{\sin \frac{q_1 - p}{2}} \frac{e^{i(n-n_1+1/2)(q_2-p)}}{\sin \frac{q_2 - p}{2}} . \qquad (31)$$

The hydrodynamic limit of (31) can be obtained in a similar fashion as was done for the magnetization profile. Expanding around the poles of the integrand and using the properties of the $\Theta$ function (see Appendix C for details) one obtains

$$\mathcal{C}_{m,n}(t) \simeq 1 - 2 \int_{-\pi}^{\pi} \frac{dP}{2\pi} \Theta(v_P - \mu) \Theta(v - v_P), \qquad (32)$$

where the ray variables

$$\mu = \frac{m - n_1 + 1/2}{t}, \qquad v = \frac{n - n_1 + 1/2}{t} \qquad (33)$$

are measured from the initial domain wall location and the expression has a very simple interpretation. Let us assume $v > \mu$ and consider the contribution of a single quasi-particle traveling at speed $v_P$. Now, for short times $v_P < \mu$ the excitation has not yet reached the first spin and thus the correlations are ferromagnetic. Once $\mu < v_P < v$, the first spin has been flipped while the second one is still untouched, hence the correlation is antiferromagnetic.

Finally, after the excitation has traveled through, $v_P > v$, the second spin is also flipped and the correlation becomes ferromagnetic again.

It turns out that, instead of approximating the integrals in (31), there is a way to directly relate $\mathcal{C}_{m,n}(t)$ to the profile $\mathcal{M}_n(t)$. Indeed, by turning the integral over $p$ into a contour integral and applying the residue theorem, one obtains the formula (80) reported in Appendix C, which is an exact relation at the level of one-particle form factors. However it is easy to see that, similarly to the hydrodynamic approximation in (32), it yields perfect ferromagnetic correlations $\mathcal{C}_{m,n}(t) \simeq 1$ when both spins are outside the front region. Indeed, it can be shown that the many-particle form factors are the ones responsible for the exponentially decaying correlations $\mathcal{C}^0_{m,n}$ in the ground state [33]. One can thus reincorporate these correlations into the approximation as

$$\mathcal{C}_{m,n}(t) \simeq \mathcal{C}^0_{m,n} + \mathcal{M}_m(t) - \mathcal{M}_n(t). \tag{34}$$

The relation in (34) is tested against exact numerical calculations for the TI chain in Fig. 5. We have calculated the correlations along the front region while keeping the distance $d$ between the spins fixed. One can see that, for $d = 1$, there is still a slight deviation from (34) which, however, decreases with increasing $d$. For $d = 10$ one has already an excellent agreement with no visible deviations. In fact, for $|n - m| \gg \xi$ one has $\mathcal{C}^0_{m,n} \to 1$, and one recovers the one-particle result (80) which should become exact. Note, however, that calculating the corrections to (34) would require to evaluate multiple integrals with higher-order offdiagonal form factors and is thus a difficult task. Nevertheless, a closer investigation of the form-factor structure in (48) confirms, that the dominant pole contribution is suppressed and thus one indeed obtains subleading terms.

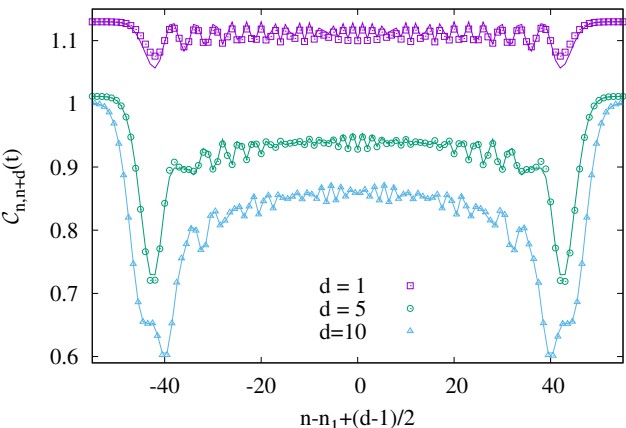

Figure 5: Equal-time correlation functions at $t = 50$ for the TI model at $h = 0.9$, for various distances $d$ between the spins. The solid lines show the approximation in (34).

## 5 Entanglement dynamics

So far we have studied the simplest observables. One can, however, gather important information about the time-evolved state by looking at the entanglement dynamics. In particular, we are interested in the entanglement profiles along the front region, considering a bipartition into two disjoint segments $A = [1, N/2 + r]$ and its complement $B$, and calculating the resulting von Neumann entropy. Entanglement profiles for domain-wall type initial conditions have been

studied extensively for time evolution under critical Hamiltonians [13, 15, 17, 22, 24, 46–49], and even a description in terms of CFT has been given [50, 51]. However, much less is known about the non-critical case, such as the one at hand.

The calculation of the entanglement profile is straightforward in MPS calculations, however, extracting the entropy via covariance-matrix techniques for Gaussian states [52, 53] requires some extra considerations. Indeed, the problem lies in the nature of the initial state, since the excitations are created upon the symmetry-broken ground state, which is inherently non-Gaussian [54]. Nevertheless, this difficulty can be overcome by relating the problem to the one where the very same excitations are created upon the Gaussian, non-magnetized ground states in (8). The method has already been outlined in [26] but we expand here the arguments for completeness.

Let us consider initial states corresponding to the two symmetry-broken ground states of the system. Using (8), the density matrices are given by

$$|\Uparrow\rangle\langle\Uparrow| = \rho_e + \rho_o\,, \qquad |\Downarrow\rangle\langle\Downarrow| = \rho_e - \rho_o\,, \tag{35}$$

where the even and odd parity components, satisfying $[\mathcal{P}, \rho_e] = 0$ and $\{\mathcal{P}, \rho_o\} = 0$, respectively, are defined as

$$\rho_e = \frac{1}{2}\big(|0\rangle_{NS\ NS}\langle 0| + |0\rangle_{R\ R}\langle 0|\big)\,, \qquad \rho_o = \frac{1}{2}\big(|0\rangle_{NS\ R}\langle 0| + |0\rangle_{R\ NS}\langle 0|\big)\,. \tag{36}$$

Clearly, the problem is with the odd component $\rho_o$, since a Gaussian density operator is by definition even. One can, however, eliminate $\rho_o$ by considering an equal-weight convex combination of the density matrices in (35). The resulting density matrix $\rho_e$ is itself still a convex combination of two Gaussian states from the NS and R sectors. However, working in the thermodynamic limit, these two states become indistinguishable [54], and one concludes that $\rho_e$ is equivalent to a proper Gaussian state.

Furthermore, as shown in Ref. [55], excitations that can be written as a product of Majorana fermions

$$D_J = \prod_{j\in J} a_j\,, \tag{37}$$

where $J$ is an arbitrary index set, preserve Gaussianity. So does unitary time evolution governed by a quadratic Hamiltonian. Hence, introducing the notation

$$\rho_A^{\Uparrow} = \mathrm{Tr}_B\big[e^{-iHt} D_J |\Uparrow\rangle\langle\Uparrow| D_J^\dagger e^{iHt}\big]\,, \qquad \rho_A^{\Downarrow} = \mathrm{Tr}_B\big[e^{-iHt} D_J |\Downarrow\rangle\langle\Downarrow| D_J^\dagger e^{iHt}\big]\,, \tag{38}$$

for the *reduced* density matrices of a given bipartition, after exciting and time evolving the initial states in (35), we finally come to the conclusion that

$$\rho_A = \frac{\rho_A^{\Uparrow} + \rho_A^{\Downarrow}}{2} \tag{39}$$

is a well-defined Gaussian state living on the Hilbert space of segment $A$.

Our goal is now to relate the entropy $S(\rho_A^{\Uparrow}) = -\mathrm{Tr}\,\rho_A^{\Uparrow} \ln \rho_A^{\Uparrow}$ of our target state to that $S(\rho_A)$ of the Gaussian state in (39). To this end we use the inequality for convex combinations of density matrices [56, 57]

$$S\Big(\sum_i \lambda_i \rho_i\Big) \le \sum_i \lambda_i S(\rho_i) - \sum_i \lambda_i \ln \lambda_i\,. \tag{40}$$

First, we note that from trivial symmetry arguments one has $S(\rho_A^{\Downarrow}) = S(\rho_A^{\Uparrow})$. Furthermore, it is also known [57] that the inequality (40) is saturated if the ranges of $\rho_i$ are pairwise orthogonal, which is again clearly satisfied in our case due to $\langle\Uparrow | \Downarrow\rangle = 0$. Hence one finds

$$S(\rho_A^{\Uparrow}) = S(\rho_A) - \ln 2\,. \tag{41}$$

Finally, it remains to calculate the covariance matrix $\Gamma_A$ corresponding to $\rho_A$, from which the calculation of the entropy $S(\rho_A)$ follows standard procedure [52, 53]. Since $\rho_A$ is the reduced density matrix of the time-evolved and excited ground state $|\psi_t\rangle_{NS}$, $\Gamma_A$ is just the reduction of the full covariance matrix with elements $\Gamma_{k,l} = {}_{NS}\langle\psi_t|[a_k, a_l]|\psi_t\rangle_{NS}/2$. This can be obtained by working in the Heisenberg picture. Since $D_J$ is unitary, $D_J D_J^\dagger = \mathbb{1}$, the effect of the excitation can be absorbed by a change of the Majorana basis [55]

$$a'_k = D_J^\dagger a_k D_J = \sum_{l=1}^{2N} Q_{k,l} a_l. \tag{42}$$

The orthogonal transformation $Q$ has a simple diagonal matrix form

$$Q_{kl} = \delta_{k,l} \prod_{j \in J} (2\delta_{k,j} - 1), \tag{43}$$

with entries $\pm 1$, depending on whether the corresponding column is part of the index set $J$ or not. In complete analogy, the unitary time evolution corresponds to the basis rotation

$$a'_k(t) = e^{iHt} a'_k e^{-iHt} = \sum_{l=1}^{2N} R_{k,l} a'_l, \tag{44}$$

where the explicit form of the orthogonal matrix $R$ was reported in Ref. [25]. Putting everything together, one finds that

$$\Gamma = R Q \Gamma_0 Q^T R^T, \tag{45}$$

where $\Gamma_0$ is the ground-state covariance matrix with elements $(\Gamma_0)_{k,l} = {}_{NS}\langle 0|[a_k, a_l]|0\rangle_{NS}/2$.

We are now ready to discuss the entanglement dynamics for the simple excitations introduced in Sec. 3. In each case we have verified that the entropy obtained by the procedure outlined above agrees perfectly with the results of our MPS calculations.

## 5.1 Single domain wall

The entropy profiles for the single domain wall, located initially in the center ($r = 0$) of the chain, have already been considered in [26] and are shown in the left of Fig. 6 for $\gamma = 0.5$ and several values of $h$. The profile $\Delta S(r) = S(\rho_A^\Uparrow) - S_0$ is always measured from the initial entropy $S_0$ of the bulk ferromagnetic state, and is plotted against the rescaled distance $\zeta = r/t$ from the center of the chain. The main feature to be seen is the emergence of a kink in the profile for $h < h_c$, at the value $\zeta_*$ that equals the local maximum of the quasiparticle velocity, in complete analogy to the case of the magnetization.

Due to the similar features observed in the entropy and magnetization profiles, one is naturally led to the question whether there is a simple relation between the two of them. We are also motivated by recent results of Refs. [27, 28], where the entanglement content of particle excitations in $1 + 1$-dimensional massive quantum field theories was studied, with a surprisingly simple result. Namely, it has been found that the entropy difference (relative to the ground state) of a single-mode excitation is independent of the wavenumber and given by the binary entropy formula involving the ratio of the subsystem and full system lengths [27, 28]. This ratio is just the density fraction of the single-mode excitation that is contained within the subsystem.

Inspired by these findings, we put forward the following ansatz

$$\Delta S(\zeta) = -\mathcal{N}\ln\mathcal{N} - (1-\mathcal{N})\ln(1-\mathcal{N}), \qquad \mathcal{N}(\zeta) = \int_{-\pi}^{\pi} \frac{dP}{2\pi}\Theta(v_P - \zeta). \tag{46}$$

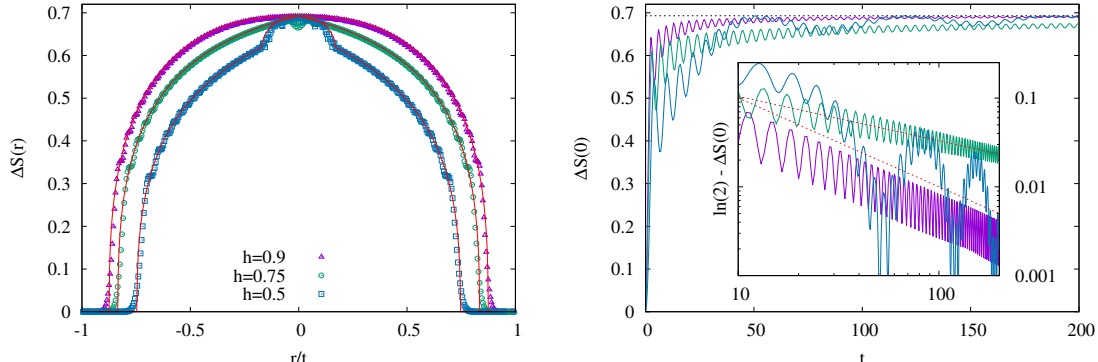

Figure 6: Left: entanglement profiles for the single domain wall, for various $h$ and $\gamma = 0.5$. The parameters are $N = 400$, $n_1 = 201$ and $t = 200$. The solid red lines show the ansatz (46). Right: half-chain entanglement as a function of time. The horizontal dotted line indicates the value $\ln 2$. The inset shows the deviation from $\ln 2$ on a logarithmic scale. The red dashed lines with slopes $-1/2$ and $-1$, respectively, are guides to the eye.

In other words, we assume that the static results of [27, 28] would generalize to our dynamical scenario, and the entropy difference for bipartitions along the ray $\zeta$ is just given by the same binary formula, with the density ratio $\mathcal{N}(\zeta)$ being the fraction of the quasiparticles that have reached the entangling point. Surprisingly, we find that the simple-minded ansatz (46), shown by the red solid lines in the left of Fig. 6, gives a very good description of the entropy profiles. Via the density fraction $\mathcal{N}(\zeta)$, the entropy profiles are thus directly related to those of the magnetization (15).

In case $h < h_c$, one observes some deviations from the ansatz (46), which are only visible in the regime $\zeta < \zeta_*$ and are assumed to be finite-time effects. In order to better understand the convergence, on the right of Fig. 6 we also studied the time evolution of the half-chain entropy $\Delta S(0)$, for the same parameter values. Although each of them can be seen to converge towards the asymptotic value $\ln 2$, their approach is rather different. For $h > h_c$ the convergence is fast and steady, with rapid oscillations only, whereas for $h < h_c$ there is a smaller frequency appearing with a larger amplitude, and the curve bounces back from its asymptotical value repeatedly. Interestingly, at the critical point $h = h_c = 0.75$ one can see a slowing down in the convergence, which becomes most evident on a logarithmic scale as shown on the inset of the figure. Indeed, the approach seems to be a power law $t^{-1/2}$, as opposed to $t^{-1}$ in the $h \neq h_c$ case. This critical slowing down is responsible for the dip around $\zeta = 0$ in the profile for $h = h_c$ on the left of Fig. 6.

One should stress the marked difference of the entropy profiles as compared to domain-wall evolution in critical systems, such as the XX chain. Indeed, in the latter case the entropy was found to grow logarithmically in time in the entire front region [47, 51], whereas here the profiles converge to the scaling function (46) when plotted against $\zeta = r/t$. In particular, the result $\Delta S(0) = \ln 2$ for $\zeta = 0$ implies that the entropy converges towards the value attained in the ground state $|0\rangle_{NS}$, which has been studied in [58, 59]. Indeed, applying the relation (41) at $t = 0$, one finds that the entropy $S_0$ in the initial symmetry-broken ground state is exactly $\ln 2$ less than that of the NS ground state. This strongly suggests that the steady state is nothing but the ground state with its symmetry restored.

## 5.2 Double domain wall

The profiles for the double domain wall are shown in Fig. 8 for various times and two different model parameters. In both cases, the profiles resemble those of two separate single domain walls for short times, while for large times the main feature is the emergence of an additional plateau in the overlap region. This strongly suggests the relation

$$\Delta S_{n_1, n_2}(r) = \Delta S_{n_1}(r) + \Delta S_{n_2}(r), \tag{47}$$

where $\Delta S_{n_1, n_2}(r)$ and $\Delta S_{n_i}(r)$ denote the entropy differences for double and single domain walls, respectively, with the indices referring to the initial locations of the excitations. In other words, one expects the entropy differences to be additive, which is indeed perfectly confirmed by the numerics.

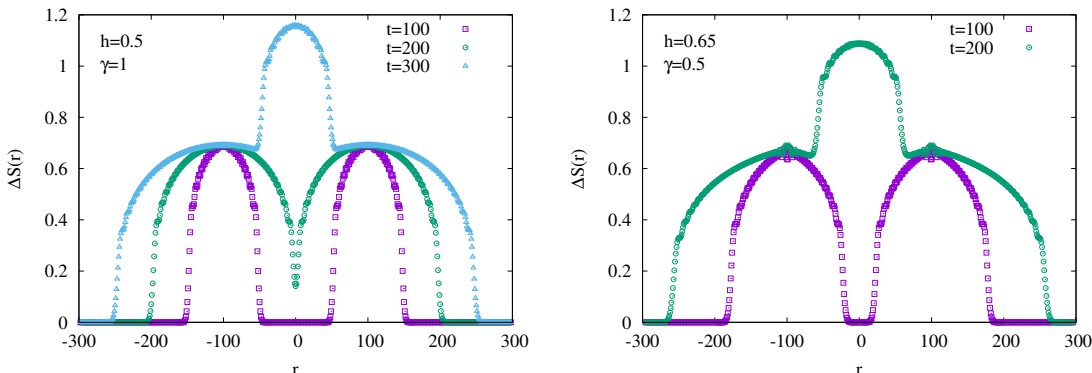

Figure 7: Entanglement profiles after the double domain wall excitation for different $h$ and $\gamma$. The parameters are the same as in Fig. 1.

## 5.3 Single spin-flip

Finally, we consider the entropy profiles for the spin-flip excitation, with the results shown in Fig. 8, for the same choice of parameters as for the magnetization profiles in Fig. 2. When plotted against the scaling variable $\zeta$, the profiles show a different behaviour as compared to those of the single domain wall excitation in Fig. 6. In particular, the additivity (47) is not satisfied, analogously to the corresponding result (25) for the magnetization, which does not have a factorized form. Indeed, as explained under Sec. 3.3, this has to do with an interference effect in the dynamics, where an incoming momentum of the first excitation can travel forward as an outgoing momentum of the second one. Clearly, such a process creates entanglement between the quasiparticles building up the two domain-wall excitations, which spoils the additivity and reduces the overall entropy of the state. Unfortunately, despite the qualitative understanding of the origin of the nontrivial entropy behaviour, we have not been able to find an ansatz analogous to (46) that captures the profiles quantitatively.

## 6 Discussion

We studied the time evolution of the magnetization and entanglement profiles in the XY chain for simple initial states that can be written as a product of one or two local fermionic excitations. The former corresponds to a single domain wall in the spin-picture and the magnetization profile has a simple hydrodynamic limit (15), corresponding to the motion of independent quasiparticles. Furthermore, in the very same limit we find that the entropy is given by the

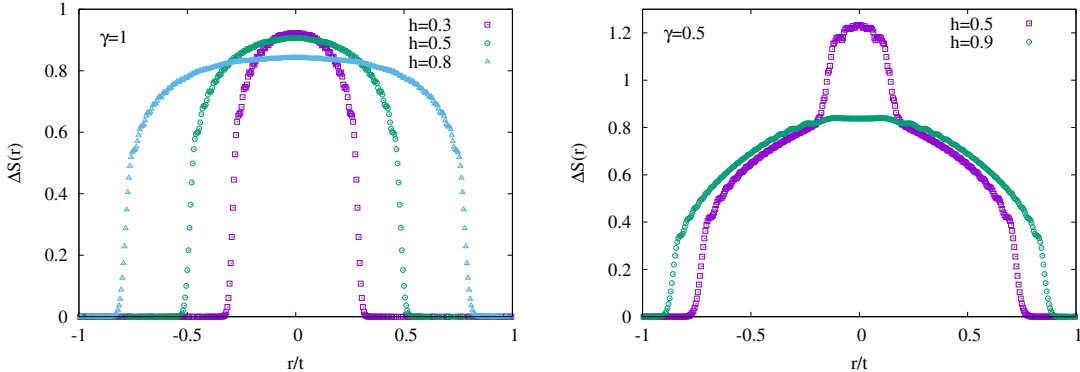

Figure 8: Entanglement profiles after a spin-flip excitation. The parameters are the same as in Fig. 2.

simple ansatz (46) and is thus directly related to the magnetization profile. The correlation function is also found to be related via (29) to the magnetization, which gives a very good approximation even for finite times and distances.

For double domain walls, excited by the product of two fermions separated by a large distance, we simply find the factorized form (22) for the magnetization, as well as the additivity (47) of the entropy differences. For a single spin-flip, however, the fermions are located on neighbouring sites and the excitation cannot be considered strictly local any more. As a consequence, we find an interference term in the magnetization profile (25). Furthermore, the additivity of the entropy is lost, and we find convergence towards a nontrivial profile

We have also compared the profiles for the single domain wall to the ones obtained via a local cut and glue quench, where the two ferromagnetic ground states are prepared on half-chains and joined together. Rather surprisingly we found that, while for the TI chain the respective profiles almost coincide, for the generic XY case they become completely different (see Fig. 4), with the discrepancy growing with the distance from the TI limit. Apparently the local quench is well approximated by a single fermionic excitation for the TI but not any more for the XY case. The precise origin of this phenomenon is unclear to us and requires further studies.

It would be also interesting to see if a QFT treatment of the entropy increase could be given. Even though our ansatz (46) was inspired by QFT results [27,28] on the entanglement content of particle excitations, those particles are single wave modes and there is no dynamics involved. On the other hand, for the case of critical Hamiltonians there exists a CFT framework for calculating the time evolution of entropy after spatially local excitations [60]. Whether this approach could be generalized to a massive QFT to predict the asymptotic entropy increase after the excitations considered in this paper is a puzzling question to be addressed.

One could also think about extending the studies to excitations composed of a product of more than two fermions. While being a straightforward generalization, the form-factor calculations are likely to be very cumbersome, due to the increasing number of the pole contributions one has to account for. Finally, it is natural to ask how the setup could be extended to interacting integrable systems, and if the treatment of such composite but still essentially local excitations could be incorporated into the theory of GHD.

## Acknowledgements

We thank H. G. Evertz for discussions and collaboration on a related previous project.

**Funding information**  The authors acknowledge funding from the Austrian Science Fund (FWF) through Project No. P30616-N36, and through SFB ViCoM F41 (Project P04).

# A  Form factors for the TI and XY chains

Here we present the form factors used in the calculations of the main text. Although for our simple excitations we required only few-particle form factors, the general expression is reported for completeness. The formula is rather involved even after taking the thermodynamic limit $N \to \infty$, and for the TI chain ($\gamma = 1$) it reads [32]

$$
\frac{{}_R\langle p_1,\ldots,p_L|\sigma_n^x|q_1,\ldots,q_K\rangle_{NS}}{{}_R\langle 0|\sigma_n^x|0\rangle_{NS}} = i^{-(K+L)/2}(-1)^{L(L-1)/2} h^{(K-L)^2/4} e^{in(\sum_{k=1}^K q_k - \sum_{l=1}^L p_l)}
$$

$$
\times \prod_{k=1}^K \frac{1}{\sqrt{N\epsilon_{q_k}}} \prod_{l=1}^L \frac{1}{\sqrt{N\epsilon_{p_l}}} \prod_{k<k'=1}^K \frac{\sin\frac{q_k-q_{k'}}{2}}{\frac{\epsilon_{q_k}+\epsilon_{q_{k'}}}{2}} \prod_{l<l'=1}^L \frac{\sin\frac{p_l-p_{l'}}{2}}{\frac{\epsilon_{p_l}+\epsilon_{p_{l'}}}{2}} \prod_{k=1}^K \prod_{l=1}^L \frac{\frac{\epsilon_{q_k}+\epsilon_{p_l}}{2}}{\sin\frac{q_k-p_l}{2}}. \tag{48}
$$

We have assumed here that the number of momenta $K$ and $L$ on the right and left hand side have the same parity, otherwise the form factor vanishes. Note that we have normalized with the vacuum form factor, i.e. with the expectation value of the ground-state magnetization. For $K = L$ the form factors (48) depend only on the dispersion relation $\epsilon_q$, given in Eq. (6), and the values of the momenta.

For the generic case of the XY chain, the expressions become even more complicated. In the limit $N \to \infty$ they can be written as [33, 34]

$$
\frac{{}_R\langle p_1,\ldots,p_L|\sigma_n^x|q_1,\ldots,q_K\rangle_{NS}}{{}_R\langle 0|\sigma_n^x|0\rangle_{NS}} = i^{-(K+L)/2}(-1)^{L(L-1)/2} g^{(K-L)^2/4} e^{in(\sum_{k=1}^K q_k - \sum_{l=1}^L p_l)}
$$

$$
\times \cosh\frac{\sum_{k=1}^K \Delta_{q_k} - \sum_{l=1}^L \Delta_{p_l}}{2} \prod_{k=1}^K \frac{1}{\sqrt{N\sinh\Delta_{q_k}}} \prod_{l=1}^L \frac{1}{\sqrt{N\sinh\Delta_{p_l}}}
$$

$$
\times \prod_{k<k'=1}^K \frac{\sin\frac{q_k-q_{k'}}{2}}{\sinh\frac{\Delta_{q_k}+\Delta_{q_{k'}}}{2}} \prod_{l<l'=1}^L \frac{\sin\frac{p_l-p_{l'}}{2}}{\sinh\frac{\Delta_{p_l}+\Delta_{p_{l'}}}{2}} \prod_{k=1}^K \prod_{l=1}^L \frac{\sinh\frac{\Delta_{q_k}+\Delta_{p_l}}{2}}{\sin\frac{q_k-p_l}{2}}, \tag{49}
$$

where we have defined

$$
\sinh\Delta_q = \frac{\sqrt{1-\gamma^2}}{\gamma\sqrt{\gamma^2+h^2-1}}\epsilon_q, \qquad g = \frac{1-\gamma^2}{\gamma\sqrt{\gamma^2+h^2-1}}. \tag{50}
$$

The above definition is valid in the parameter regime $\sqrt{1-\gamma^2} < h < 1$, i.e. in the non-oscillatory ferromagnetic phase. In the oscillatory phase $0 < h < \sqrt{1-\gamma^2}$ the corresponding expressions can be obtained by analytic continuation [33]. One can also check that, in the singular TI limit $\gamma \to 1$, the expression (49) goes over to the one in (48). While in general they differ in the details, these will turn out to be irrelevant for the hydrodynamic limit, since their pole structure is exactly the same.

We now discuss the form factors needed in the main text. The simplest is the one-particle form factor ($K = L = 1$), where using some hyperbolic identities in (49), one can show that the TI and XY cases yield the same expression

$$
\frac{{}_R\langle p|\sigma_n^x|q\rangle_{NS}}{{}_R\langle 0|\sigma_n^x|0\rangle_{NS}} = -\frac{i}{N}\frac{\epsilon_p+\epsilon_q}{2\sqrt{\epsilon_p\epsilon_q}}\frac{e^{in(q-p)}}{\sin\frac{q-p}{2}}. \tag{51}
$$

Thus the formula (14) for the single domain wall excitation is valid for arbitrary parameter values of the XY chain. In general, no such simplification occurs and in the following we restrict ourselves to the TI case for the sake of simplicity. For the spin-flip excitation one needs the off-diagonal form factor with $K = 2$ and $L = 0$ which reads

$$\frac{_\mathrm{R}\langle 0|\sigma_n^x |q_1, q_2\rangle_\mathrm{NS}}{_\mathrm{R}\langle 0|\sigma_n^x |0\rangle_\mathrm{NS}} = -\frac{i}{N}\frac{h}{\sqrt{\epsilon_{q_1}\epsilon_{q_2}}}e^{in(q_1+q_2)}\frac{2\sin\frac{q_1-q_2}{2}}{\epsilon_{q_1}+\epsilon_{q_2}}. \tag{52}$$

One can see immediately, that this form factor does not have any poles which implies that it will only give a subleading contribution. The diagonal two-particle form factor ($K = L = 2$), on the other hand, has the form

$$\begin{aligned}
\frac{_\mathrm{R}\langle p_1, p_2|\sigma_n^x |q_1, q_2\rangle_\mathrm{NS}}{_\mathrm{R}\langle 0|\sigma_n^x |0\rangle_\mathrm{NS}} = {} & \frac{1}{N^2}\frac{e^{in(q_1+q_2-p_1-p_2)}}{\sqrt{\epsilon_{p_1}\epsilon_{p_2}\epsilon_{q_1}\epsilon_{q_2}}}\frac{2\sin\frac{p_1-p_2}{2}}{\epsilon_{p_1}+\epsilon_{p_2}}\frac{2\sin\frac{q_1-q_2}{2}}{\epsilon_{q_1}+\epsilon_{q_2}} \\
& \times \frac{\epsilon_{q_1}+\epsilon_{p_1}}{2\sin\frac{q_1-p_1}{2}}\frac{\epsilon_{q_1}+\epsilon_{p_2}}{2\sin\frac{q_1-p_2}{2}}\frac{\epsilon_{q_2}+\epsilon_{p_1}}{2\sin\frac{q_2-p_1}{2}}\frac{\epsilon_{q_2}+\epsilon_{p_2}}{2\sin\frac{q_2-p_2}{2}},
\end{aligned} \tag{53}$$

with two possible poles for $q_1 = p_1$ and $q_2 = p_2$, or with an exchange of momenta for $q_1 = p_2$ and $q_2 = p_1$. It should be noted that, for the generic diagonal $K$-particle form factors in (48), an arbitrary permutation between the incoming and outgoing momenta yields a pole, which makes the analysis of the contributions increasingly complicated.

## B  Stationary phase calculations for the profile

In this appendix we summarize the calculations leading to the approximations of the magnetization profile in the hydrodynamic regime. The simplest case is the single domain wall, where $\mathcal{M}_n(t)$ is given by a double integral (14). The integrand has a pole due to the form factor, which can be regularized as

$$\mathcal{M}_n(t) = 1 + \int_{-\pi}^{\pi}\frac{dp}{2\pi}\int_{-\pi}^{\pi}\frac{dq}{2\pi}\frac{\epsilon_p + \epsilon_q}{2\sqrt{\epsilon_p\epsilon_q}}\frac{e^{i(n-n_1+1/2)(q-p)}}{i\sin\left(\frac{q-p+i\varepsilon}{2}\right)}e^{i(\theta_q-\theta_p)/2}e^{-i(\epsilon_q-\epsilon_p)t}, \tag{54}$$

by introducing the infinitesimal shift $\varepsilon > 0$. The integrand of (54) is highly oscillatory for $|n-n_1| \gg 1$ and $t \gg 1$, and the location of the pole at $q = p$ suggests the change of variables $Q = q - p$ and $P = (q + p)/2$. The phase factors become stationary at $Q = 0$, thus the integrand should be expanded around this value. Keeping only the most singular term one has

$$1 + 2\int_{-\pi}^{\pi}\frac{dP}{2\pi}\int_{-\infty}^{\infty}\frac{dQ}{2\pi i}\frac{e^{i(n-n_1+1/2+\theta'_P-v_P t)Q}}{Q + i\varepsilon}, \tag{55}$$

where we have extended the integration in the relative momentum up to infinity. Thanks to the definition (6), the function $\theta'_P$ varies smoothly and one can neglect it in the hydrodynamic regime. Then using the integral representation of the Heaviside theta function

$$\Theta(x) = -\lim_{\varepsilon\to 0}\int_{-\infty}^{\infty}\frac{dQ}{2\pi i}\frac{e^{-iQx}}{Q + i\varepsilon}, \tag{56}$$

and introducing the ray variable $v = (n-n_1+1/2)/t$ brings us to the result (15) in the main text.

The bulk hydrodynamic profile is thus recovered by solving the equation $v_q = v$. Special attention is needed around the maximum $v_{q_*} = v_{max}$ of the velocities, where the solutions

coalesce at momentum $q_*$. To get the fine structure of the front edge, one has to expand the dispersion around $q_*$ as

$$\epsilon_q \approx \epsilon_{q_*} + v_{q_*}(q - q_*) + \frac{\epsilon'''_{q_*}}{6}(q - q_*)^3. \tag{57}$$

Furthermore, one can introduce the following rescaled variables

$$X = \left(\frac{-2}{\epsilon'''_{q_*}t}\right)^{1/3}(n - n_1 + 1/2 + \theta'_{q_*}/2 - v_{q_*}t),$$

$$Q = \left(\frac{-2}{\epsilon'''_{q_*}t}\right)^{-1/3}(q - q_*), \quad P = \left(\frac{-2}{\epsilon'''_{q_*}t}\right)^{-1/3}(p - q_*). \tag{58}$$

Substituting (57) and (58) into (54), one arrives at the following integral

$$1 + 2\left(\frac{-2}{\epsilon'''_{q_*}t}\right)^{1/3}\int\frac{dP}{2\pi}\int\frac{dQ}{2\pi}\frac{e^{iX(Q-P)}e^{i(Q^3-P^3)/3}}{i(Q-P+i\varepsilon)}. \tag{59}$$

Using the integral representation of the Airy kernel [39]

$$\mathcal{K}_{Ai}(X,Y) = \lim_{\varepsilon\to 0}\int\frac{dP}{2\pi}\int\frac{dQ}{2\pi}\frac{e^{-iXP}e^{-iP^3/3}e^{iYQ}e^{iQ^3/3}}{i(P-Q-i\varepsilon)} = \frac{Ai(X)Ai'(Y)-Ai'(X)Ai(Y)}{X-Y}, \tag{60}$$

one recovers (16) of the main text, with $\rho(X) = \lim_{Y\to X}\mathcal{K}_{Ai}(X,Y)$ given by the diagonal terms of the Airy kernel.

The hydrodynamic limit (22) for the double domain wall can be obtained in a similar fashion, however, one has now a quadruple integral to start with. The poles are contained in the two-particle form factor (53). First, we consider the pole with $q_1 = p_1$ and $q_2 = p_2$. Changing variables as

$$Q_i = q_i - p_i, \qquad P_i = \frac{q_i + p_i}{2}, \tag{61}$$

and expanding the phases around the stationary points $Q_i = 0$, one has

$$\mathcal{I}_1 = 4\int\frac{dP_1}{2\pi}\int\frac{dP_2}{2\pi}f(P_1,P_2,Q_1,Q_2)\int\frac{dQ_1}{2\pi}\frac{e^{-ix_1Q_1}}{Q_1}\int\frac{dQ_2}{2\pi}\frac{e^{-ix_2Q_2}}{Q_2}, \tag{62}$$

where we defined

$$x_i = v_{P_i}t - (-1)^i\theta'_{P_i} - (n - n_i + 1/2). \tag{63}$$

The function $f$ in (62) describes the slowly varying part of the form factor in (53). It is easy to see, that the terms containing the dispersion $\epsilon_{q_i}$ and $\epsilon_{p_i}$ can be approximated by 1 to leading order. It remains to analyze the contribution of the trigonometric factors that do not contain the poles, which can be rewritten as

$$f(P_1,P_2,Q_1,Q_2) \approx -\frac{\cos(\frac{Q_1-Q_2}{2}) - \cos(P_1-P_2)}{\cos(\frac{Q_1+Q_2}{2}) - \cos(P_1-P_2)}. \tag{64}$$

Thus, again to leading order around $Q_i = 0$, one has $f(P_1,P_2,Q_1,Q_2) \approx -1 + \mathcal{O}(Q_1Q_2)$, meaning that the first correction would already remove the singularity in the integral (62), and can be neglected. Setting $f = -1$, one recovers immediately the factorized result (22).

The second pole of the form factor (53) is given by $q_1 = p_2$ and $q_2 = p_1$ and corresponds to an exchange of the outgoing momenta. The form factor itself transforms trivially under

this exchange, acquiring only a sign. The time-evolved state (20), however, has phase factors attached to the locations of the domain walls and thus transforms nontrivially under exchange of the momenta. Indeed, introducing the variables

$$Q'_1 = q_1 - p_2, \qquad Q'_2 = q_2 - p_1, \qquad P'_1 = \frac{q_1 + p_2}{2}, \qquad P'_2 = \frac{q_2 + p_1}{2}, \qquad (65)$$

this phase factor can now be rewritten as

$$e^{-i(Q'_1 + Q'_2)(n_1 + n_2)/2} e^{i(P'_1 - P'_2)(n_2 - n_1)}. \qquad (66)$$

The second term contains the center of mass momenta and becomes highly oscillatory for $|n_2 - n_1| \gg 1$. This phase, however, cannot be made stationary, since the time-dependent part of the phase in (20) is symmetric under the exchange of the momenta. One thus concludes that, for large separations of the domain walls, the second pole gives a negligible contribution.

The situation for the spin-flip excitation is different. As discussed in the main text, except for a sign change of the Bogoliubov angles, the state (23) is a double domain wall with $n_2 = n_1 + 1$. The first pole thus yields the very same factorized result as in (62), with the corresponding changes in $x_i$. In the hydrodynamic limit, however, it is more natural to measure distances from the spin-flip location $n_1$ (instead of $n_1 \pm 1/2$) and use the ray variable $\tilde{v} = \frac{n - n_1}{t}$, which gives the second term in (25). The second pole, however, has also a significant contribution, since $n_2 - n_1 = 1$ and the phase factor in (66) now varies slowly. Expanding around $Q'_i = 0$, one finds

$$\mathcal{I}_2 = 4 \int \frac{dP'_1}{2\pi} \int \frac{dP'_2}{2\pi} e^{iP'_1} e^{i\theta_{P'_1}} e^{-iP'_2} e^{-i\theta_{P'_2}} \int \frac{dQ'_1}{2\pi} \frac{e^{-ix'_1 Q'_1}}{Q'_1} \int \frac{dQ'_2}{2\pi} \frac{e^{-ix'_2 Q'_2}}{Q'_2}, \qquad (67)$$

where $x'_i = v_{P'_i} t - (n - n_1)$ and the sign change in the form factor has been taken into account. It is easy to see that

$$\mathcal{I}_2 = - \left| 2 \int \frac{dP'}{2\pi} e^{iP'} e^{i\theta_{P'}} \int \frac{dQ'}{2\pi i} \frac{e^{-ix'Q'}}{Q'} \right|^2. \qquad (68)$$

Regularizing the pole via the identity $Q'^{-1} = i\pi\delta(Q') + \lim_{\varepsilon \to 0}(Q' + i\varepsilon)^{-1}$, using (56) and the expression of the transverse magnetization in (24), the third term of (25) follows.

It remains to investigate the edge scaling regime for the spin-flip excitation. The second term of (25) is simply the square of the profile for a single domain wall, where the edge scaling is given by (16). To leading order, this just yields a factor 2. The situation is similar for the third term in (25) where, additionally, the phase factors in the integral must be evaluated at the momentum $q_*$ where the velocity has its maximum, $v_{q_*} = v_{max}$. Collecting the terms, one obtains the prefactor in (27).

Finally it should be noted that, although the calculation above has been carried out using the form factors for the TI chain, the result generalizes to the XY case. Indeed, the pole structure of the form factors is exactly the same, whereas the differences in the slowly varying part are irrelevant in the hydrodynamic limit, since they have the same trivial limit after expanding around the pole.

## C   Calculation of correlation functions

At one-particle level of the form-factor expansion, the normalized correlation function is given by the triple integral

$$
\begin{aligned}
\mathcal{C}_{m,n}(t) \simeq \int \frac{dq_1}{2\pi} \int \frac{dq_2}{2\pi} & e^{-i(\theta_{q_1}-\theta_{q_2})/2} e^{i(\epsilon_{q_1}-\epsilon_{q_2})t} \\
& \times \int \frac{dp}{2\pi} \frac{\epsilon_p + \epsilon_{q_1}}{2\sqrt{\epsilon_p \epsilon_{q_1}}} \frac{\epsilon_p + \epsilon_{q_2}}{2\sqrt{\epsilon_p \epsilon_{q_2}}} \frac{e^{-i(m-n_1+1/2)(q_1-p)}}{\sin \frac{q_1-p}{2}} \frac{e^{i(n-n_1+1/2)(q_2-p)}}{\sin \frac{q_2-p}{2}} .
\end{aligned}
\tag{69}
$$

The stationary phase approximation of this integral is very similar to that of the magnetization profile. Introducing the new set of variables

$$
Q_1 = q_1 - p, \qquad Q_2 = q_2 - p, \qquad P = \frac{q_1 + p}{2},
\tag{70}
$$

and expanding around the poles $Q_1 = 0$ and $Q_2 = 0$, one obtains

$$
\mathcal{C}_{m,n}(t) \simeq 4 \int \frac{dP}{2\pi} \int \frac{dQ_1}{2\pi} \frac{e^{-i(m-n_1+1/2+\theta'_P-v_P t)Q_1}}{Q_1} \int \frac{dQ_2}{2\pi} \frac{e^{i(n-n_1+1/2+\theta'_P-v_P t)Q_2}}{Q_2}.
\tag{71}
$$

Applying (56) in both the $Q_1$ and $Q_2$ integrals, the result can again be written with the help of step functions

$$
\mathcal{C}_{m,n}(t) \simeq 1 - 2 \int_{-\pi}^{\pi} \frac{dP}{2\pi} [\Theta(v_P - \mu) + \Theta(v_P - \nu) - 2\Theta(v_P - \mu)\Theta(v_P - \nu)],
\tag{72}
$$

where the scaling variable $\mu = (m - n_1 + 1/2)/t$ is introduced analogously to $\nu$. Assuming $\mu < \nu$ and using the identities for the step function

$$
\Theta(v_P - \nu) = 1 - \Theta(\nu - v_P), \qquad \Theta(v_P - \mu) - \Theta(v_P - \nu) = \Theta(v_P - \mu)\Theta(\nu - v_P),
\tag{73}
$$

the result (32) of the main text follows immediately.

Instead of applying the stationary phase argument, one can also do a more precise analysis. Indeed, it turns out that the integral over $p$ in (69) can be carried out explicitly. We first regularize the factor containing the pole as

$$
\frac{1}{\sin\left(\frac{q_1-p}{2}\right)\sin\left(\frac{q_2-p}{2}\right)} = \left[2\pi\delta(p - q_1) + \frac{1}{i\sin\left(\frac{q_1-p+i\varepsilon}{2}\right)}\right]\left[2\pi\delta(p - q_2) + \frac{1}{i\sin\left(\frac{p-q_2+i\varepsilon}{2}\right)}\right].
\tag{74}
$$

Multiplying out this expression, the terms containing the delta functions can be plugged back into (69) and integrated over. Comparing to (54), one can identify the resulting double integrals as $\mathcal{M}_m(t) - 1$ and $\mathcal{M}_n(t) - 1$, respectively, while the product of the delta functions trivially yields one. The remaining factor from (74) can be rewritten as

$$
\frac{1}{\sin\left(\frac{q_1-p+i\varepsilon}{2}\right)\sin\left(\frac{p-q_2+i\varepsilon}{2}\right)} = \frac{2}{\cos\left(\frac{q_1+q_2}{2} - p\right) - \cos\left(\frac{q_1-q_2}{2} + i\varepsilon\right)}.
\tag{75}
$$

Introducing new variables

$$
z = e^{i[p-(q_1+q_2)/2]}, \qquad z_0 = e^{i[(q_1-q_2)/2+i\varepsilon]},
\tag{76}
$$

the integral over $p$ is transformed into the contour integral

$$\mathcal{I} = \oint \frac{dz}{2\pi i z} f(z) \frac{4}{z + z^{-1} - (z_0 + z_0^{-1})}, \tag{77}$$

where $f(z)$ is the slowly varying regular part of the integrand in (69), and the contour is the unit circle. Now the two poles are located at $z = z_0$ and $z = z_0^{-1}$. However, for $\varepsilon > 0$, only $z = z_0$ lies inside the contour and contributes to the integral. We have thus to obtain the residue around this pole. Rewriting

$$\frac{4}{z^2 + 1 - z(z_0 + z_0^{-1})} = \frac{4}{z_0 - z_0^{-1}} \left( \frac{1}{z - z_0} - \frac{1}{z - z_0^{-1}} \right), \tag{78}$$

and the two poles correspond to $p = q_1$ and $p = q_2$, respectively. Hence the result of the contour integral is

$$\mathcal{I} = \frac{2f(q_1)}{i \sin\left(\frac{q_1 - q_2}{2} + i\varepsilon\right)}. \tag{79}$$

Finally, noting that $\mathcal{I}$ enters with a minus sign (see (74)), and inserting the result back into (69), one can easily identify the contribution as $-2(\mathcal{M}_n(t) - 1)$. Collecting all the terms, one arrives at the result

$$\mathcal{C}_{m,n}(t) \simeq 1 + \mathcal{M}_m(t) - \mathcal{M}_n(t). \tag{80}$$

As a closing remark, we give a simple argument why the many-particle contributions in the form-factor expansion of the correlation functions can be neglected. In the one-particle expression (69), the dominant contribution is obtained from momenta satisfying $q_1 = p = q_2$, where the stationary phase conditions match the poles of the integrand. The next nonvanishing term in the expansion involves three intermediate particles, where the phase factor could be made stationary for $q_1 = p_1 = q_2$ and $p_2 = -p_3$. However, from (48) one can see that there is no pole in the form factor at $p_2 = -p_3$, and thus the contribution is subleading.

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
