# Peer review of "Front dynamics in the XY chain after local excitations"

_SciPost Physics, doi:SciPost Phys. 8, 037 (2020)_

## Round 1 · Referee Report · Anonymous (Referee 1) · 2019-10-20

Strengths

1- Potentially interesting generalizations to interacting systems 2- Clearly stated results

Weaknesses

1- Some of the results already appeared in Ref. [25] by the same authors.

Report

In this paper, the authors investigate the out of equilibrium dynamics of the XY spin chain, in a regime for which a simple hydrodynamic picture is expected. The initial states are however not standard, with domain wall created on to of the (symmetry-broken) ground states. This means that a hydrodynamic description is, a priori, not obvious.

Nevertheless, the authors use the exact solvability of the model and known form factors to compute simple observable such as magnetization. This is done by saddle point analysis. The final result sometimes admits a simple hydrodynamic description, but there can also be more complicated interference effects. Results are also obtained for the edge of the obtained hydrodynamic profiles, which are described by the Airy kernel.

Perhaps the most interesting results are for the entanglement. It is described by a simple scaling ansatz for certain protocols (such as 'double domain wall'), and checked numerically with good agreement. Other setups are left open to future analytical investigations.

Overall the paper is clear and reasonably well written. Given the timely nature of the results I recommend publication in Scipost, provided the comments below are be addressed:

1) Page 5, is the use of MPS techniques mere convenience, or are there difficulties associated to using Pfaffian techniques for more complicated correlations than magnetization?

2) The authors should be more careful about the use of the word local in several places. For instance, after (18), the operator is not local in terms of fermions in the limit considered right after (19), by any reasonable definition of local. Similar comment at the top of page 9.

3) When doing form factors expansions, only the lowest order contributions are kept. Then, the authors perform a saddle point analysis of the resulting integrals, to get hydrodynamic behavior. However, it should be possible to show that higher order form factor contributions become subleading in the long time limit. Can the authors comment on that?

4) In section 5.1, it is not really clear why massive QFT results are relevant to the problem discussed here, since massive field theories describe only the vicinity of a critical point. Perhaps there is a better justification of the binary entropy ansatz of (46).

5) Page 16, second paragraph. I do not understand the meaning of 'critical point' or 'critical slowing down' for '$h_c=0.75$'. Can the authors also comment on the use that can be made of (34)?

Here is also a list of misprints:

a) Before (15), 'In turn' should be replaced by something else.

a) End of section 3.2. 'is due the fact' should read 'is due to the fact'

b) Beginning ofsection 5. 'are interested about' should read 'are interested in'

c) After (40), what are 'the ranges' of $\rho_i$ ?

d) After (41), 'thus' should be removed

Requested changes

See report

  • validity: high
  • significance: good
  • originality: good
  • clarity: high
  • formatting: excellent
  • grammar: excellent

Author:  Viktor Eisler  on 2020-02-19  [id 743]

(in reply to Report 1 on 2019-10-20)

We thank the referee for the careful reading of the manuscript and for the insightful comments. Below we address the referee's questions and provide some clarifications, indicating the changes to the manuscript.

1) Page 5, is the use of MPS techniques mere convenience, or are there difficulties associated to using Pfaffian techniques for more complicated correlations than magnetization?

The Pfaffian technique works very well and without difficulties for both the magnetization as well as for the correlations after the excitations. The use of MPS serves rather as a benchmark of our routine in this case. However, for the local quench in Sec. 3.4 one has some subtleties when calculating the magnetization via a sum of Pfaffians, and the MPS calculations are actually more convenient. Moreover, the procedure of calculating the entropy via Gaussian techniques involves some nontrivial steps and approximations, where again MPS provided a very important crosscheck of our results, as stated just before Sec. 5.1.

2) The authors should be more careful about the use of the word local in several places. For instance, after (18), the operator is not local in terms of fermions in the limit considered right after (19), by any reasonable definition of local. Similar comment at the top of page 9.

We agree with the referee that the uniform usage of the word local might be somewhat misleading. We now introduced the term two-local whenever we are referring to fermion operators that are supported on two sites.

3) When doing form factors expansions, only the lowest order contributions are kept. Then, the authors perform a saddle point analysis of the resulting integrals, to get hydrodynamic behavior. However, it should be possible to show that higher order form factor contributions become subleading in the long time limit. Can the authors comment on that?

At one-particle level the correlation function gets a dominant contribution since the stationary phase condition $q_1=p=q_2$ matches the poles of the integrand. The first correction in the form-factor expansion involves three intermediate particles. One could still make the phase in the corresponding integrals stationary by choosing $q_1=p_1=q_2$ and $p_2=-p_3$. However, this does not match with the pole structure of the form factor. This simple argument shows why the corresponding contribution becomes subleading. We added a paragraph explaining this argument at the end of Appendix C.

4) In section 5.1, it is not really clear why massive QFT results are relevant to the problem discussed here, since massive field theories describe only the vicinity of a critical point. Perhaps there is a better justification of the binary entropy ansatz of (46).

We agree that there must be a more solid justification of the ansatz (46), e.g. starting from the covariance matrix, but we have not yet been able to find it. Nevertheless, our ansatz for the entropy was inspired by the results available from massive QFT, combined with the quasi-particle picture describing the dynamics in our lattice model.

5) Page 16, second paragraph. I do not understand the meaning of 'critical point' or 'critical slowing down' for 'h_c=0.75'. Can the authors also comment on the use that can be made of (34)?

The critical point is the one we found already in our previous work, Ref. [26] in the new version of the manuscript, and is thus only shortly discussed at the end of Sec. 3.1. At $h=h_c$ the dispersion of the model changes qualitatively, leading to a kind of phase transition in the hydrodynamic profile. The term "critical slowing down" simply refers to the fact that the approach towards the steady state (as indicated by the entropy) seems to be much slower when one is sitting at the critical point.

Formula (34) gives an approximation of the correlations in terms of the magnetization profiles, which works rather well. It essentially shows that there is not too much extra information in the correlations.

Here is also a list of misprints:

a) Before (15), 'In turn' should be replaced by something else.

a) End of section 3.2. 'is due the fact' should read 'is due to the fact'

b) Beginning ofsection 5. 'are interested about' should read 'are interested in'

c) After (40), what are 'the ranges' of $\rho_i$ ?

d) After (41), 'thus' should be removed

We have corrected the misprints. The range of an operator is just the image of its domain.

---

## Round 1 · Referee Report · Anonymous (Referee 2) · 2020-2-13

Strengths

1- the problem addressed is general 2- the analytical results are supported by numerical simulations 3- the paper is very readable

Weaknesses

1- the model is not interacting

Report

This paper reports analytical and numerical investigations into the dynamics of a particular class of "excitations" in the XY chain. The authors consider three cases (single domain wall, double domain wall, and single spin flip) in which the ground state is perturbed by simple operators that have a local representation in terms of the Jordan-Wigner fermions in terms of which the Hamiltonian is quadratic. The hydrodynamic limit of the magnetisation profile is computed exactly. The equal-time two-point function of the x component of the spin is obtained at the lowest order of a form-factor expansion. The bipartite entanglement profile is conjectured and the prediction is numerically checked using tensor network algorithms. In the final example, the authors investigate the time evolution of the aforementioned quantities after joining together the two ground states of the ferromagnetic Hamiltonian.

I think that the paper is well written and meets the criteria of acceptance of SciPost. My only criticism is that the examples considered lack a bit of courage, investigating only cases that could have been addressed by rather standard means.

Requested changes

1- In my opinion the authors should comment a bit more on whether some of their results might be strong enough to hold true in the presence of interactions.

  • validity: high
  • significance: high
  • originality: good
  • clarity: top
  • formatting: perfect
  • grammar: excellent

Author:  Viktor Eisler  on 2020-02-19  [id 742]

(in reply to Report 2 on 2020-02-13)

1- In my opinion the authors should comment a bit more on whether some of their results might be strong enough to hold true in the presence of interactions.

Unfortunately, this is a very hard question to answer. Already in the case of the XY chain, obtaining the hydrodynamic limit of the magnetization profile requires a careful analysis of the form factors and their pole structure. Although form factors are in principle available for some interacting integrable models such as the XXZ chain, their structure is even more involved and the corresponding calculations are most likely rather cumbersome. Nevertheless, our physical intuition dictates that the hydrodynamic limit after local operator excitations should carry over, e.g. in the spirit of generalized hydrodynamics, where the quasiparticle velocities become dressed. However, the precise understanding of e.g. an analogous double domain wall excitation in the XXZ chain would require an in-depth study of the problem, which was clearly beyond the scope of our present work. Without such further insight we would rather not venture to comment on the generality of our results.

---

## Round 2 · Author Response

See response to referees

---

## Round 2 · List of Changes

Minor changes, see response to referees

---

## Editorial Decision

published